# Dual-color DNA-PAINT single-particle tracking enables extended studies of membrane protein interactions

Christian Niederauer[1], Chikim Nguyen[1], Miles Wang-Henderson[1], Johannes Stein[2], Sebastian Strauss[2], Alexander Cumberworth[1], Florian Stehr[2], Ralf Jungmann[2,3], Petra Schwille[2] & Kristina A. Ganzinger[1]

DNA-PAINT based single-particle tracking (DNA-PAINT-SPT) has recently significantly enhanced observation times in in vitro SPT experiments by overcoming the constraints of fluorophore photobleaching. However, with the reported implementation, only a single target can be imaged and the technique cannot be applied straight to live cell imaging. Here we report on leveraging this technique from a proof-of-principle implementation to a useful tool for the SPT community by introducing simultaneous live cell dual-color DNA-PAINT-SPT for quantifying protein dimerization and tracking proteins in living cell membranes, demonstrating its improved performance over single-dye SPT.

Single-particle tracking (SPT) is a powerful method to investigate the orchestration of biomolecular processes at cell membranes or in reconstituted systems[1,2]. To detect and follow the molecules of interest, these are usually fluorescently labeled, with observation times and localization precision depending on label brightness and photostability. Furthermore, labels have to be conjugated with the target molecule in one-to-one stoichiometry to obtain meaningful data[3]. Due to their small size, brightness and ease of chemical addressability, organic dyes conjugated to genetically encoded protein tags are currently the preferred labeling strategy for SPT of membrane proteins[4–7]. However, observations are typically only possible for a few seconds at 20–50 nm spatial precision before the dyes photobleach[2]. Short trajectories particularly hamper quantitative studies of molecular association by multi-color SPT, as they reduce the dynamic range of these experiments and make it hard to distinguish true co-diffusion events from chance encounters.

Recently, we have demonstrated how DNA-PAINT based single-particle tracking (DNA-PAINT-SPT) can increase trajectory lengths by circumventing the limited photon budget of single dyes[8]. In DNA-PAINT-SPT, short dye-labeled DNA oligonucleotides (imager strands) transiently bind to a target-bound complementary docking strand that contains several repeating and speed-optimised sequences[9,10]. As multiple imager strands can thus bind simultaneously and are designed to exchange on a time scale similar to that of dye photobleaching, this allowed us to follow the motion of DNA-origami on a supported lipid bilayer (SLB) for minutes rather than seconds[8]. While the concept of a continuous turnover of fluorophores to circumvent photobleaching has gained traction in the field of single-molecule fluorescence[11,12], it is yet to be implemented in more complex biological samples.

Here, we introduce a motif for dual-color DNA-PAINT-SPT for measuring protein-protein interactions at the single-molecule level, and use it to reliably quantify ligand-induced protein dimerization in membranes. We further extend our dual-color DNA-PAINT-SPT implementation to live cell imaging applications and demonstrate its improved performance over single-dye SPT.

## Results

### Orthogonal docking strands for reliable detection of protein-protein interactions in dual-color SPT

To apply DNA-PAINT-SPT in dual-color experiments of molecular interactions, we designed orthogonal docking-imager strand pairs (Fig. 1a, Supplementary Note 1) that exhibited negligible crosstalk (localization densities of poly(TC)- and poly(AC)-labeled FK506

[1]Autonomous Matter Department, AMOLF, Amsterdam, The Netherlands. [2]Max Planck Institute of Biochemistry, Martinsried, Germany. [3]Faculty of Physics, Ludwig Maximilian University, Munich, Germany. ✉e-mail: k.ganzinger@amolf.nl

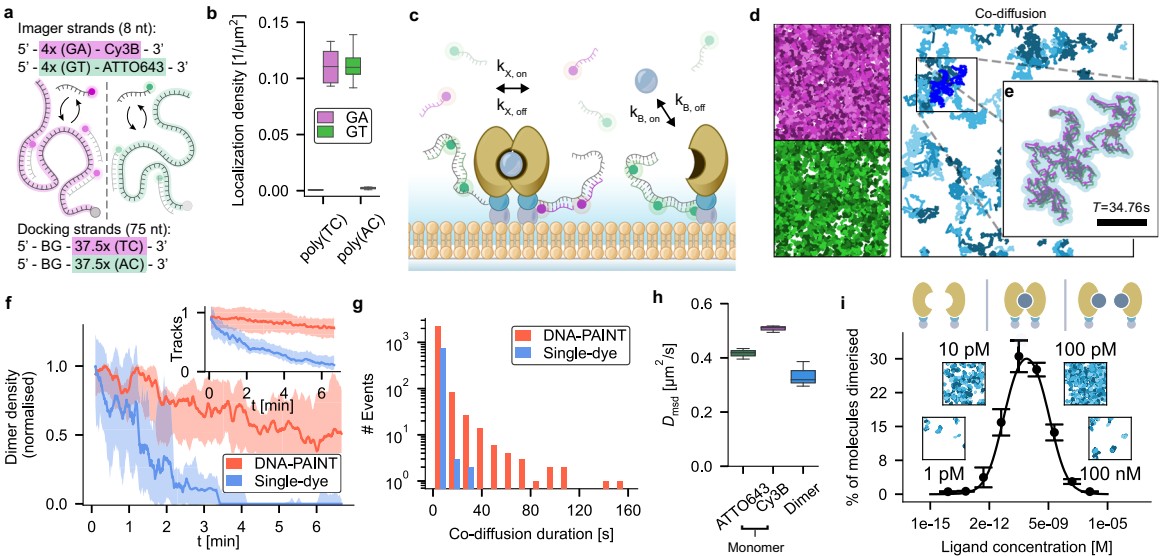

**Fig. 1 | DNA-PAINT-SPT allows quantitative single-molecule studies of FKBP homodimerization. a** Sequence design for dual-color DNA-PAINT-SPT. Imager strands are 8 nucleotide long fluorescently labeled single-stranded DNA consisting of nucleotide-pair repeats (GA and GT). Docking strands are 75 nucleotide long single-stranded DNA, consisting of repeats of the respective complementary nucleotide-pairs (TC and AC) and a benzylguanine (BG) moiety for covalent labeling of proteins with a SNAPtag. A single docking strand can be occupied by several imager strands at the same time, with binding times tuned to match bleaching kinetics, allowing for the continuous observation of labeled proteins. **b** Density of tracks of reconstituted SNAPtag fusion proteins labeled with BG-conjugated orthogonal docking strand sequences and their respective imager strands conjugated to ATTO643 or Cy3B fluorophores. Boxes, line and whiskers show, respectively, 25–75 quartiles, median, and minimum and maximum values of data pooled from three fields of view of duplicate samples per condition. **c** Schematic of the in vitro dimerization assay. FKBP-proteins (yellow) are reconstituted on a supported lipid bilayer and labeled via their SNAPtag (blue) with orthogonal DNA oligonucleotide docking strands. Complementary (8 nucleotides) imager strands, conjugated to Cy3B (magenta) and ATTO643 (green) fluorophores transiently bind to the docking strand and allow for dual-color single-particle tracking. Dimerization of FKBP-proteins is induced by adding a dimerization agent (gray sphere). **d** Single-molecule trajectories collected during a 40 s recording (left, magenta and green) and detected dimerization events (right, blue). Fields of view are 70 μm × 70 μm. This experiment was repeated independently six times with similar results. **e** Tracks of two monomers (magenta and green) co-diffusing for 34.76 s (869 frames). For displaying purposes, tracks were moved in opposite *x* and *y*-directions by 0.2 μm. Co-diffusion as detected by the tracking algorithm is presented as a blue-shaded track. Scale bar: 5 μm. **f** Mean number of dimers per frame observable using Cy3B- and ATTO643 DNA-PAINT-SPT (red) or Cy3B- and ATTO643 single-dye control (blue), normalised to their respective initial values. Inset: Number of tracks using DNA-PAINT-SPT (red) or single-dye control (blue), normalised to their respective initial values. Curves represent the median with shaded areas indicating the 25–75 quartiles of data collected from three samples per condition. **g** Histogram of interaction durations for ligand-induced dimerization as detected using Cy3B- and ATTO643 DNA-PAINT-SPT (red, mean = 24.2 ± 3.8 s) or Cy3B- and ATTO643 single-dye labeling (blue, mean = 3.3 ± 1.3 s). Data collected during measurements of 6 min each from three samples per condition. **h** Diffusion constants of monomers and dimers labeled with DNA-PAINT or single dyes. Boxes, line and whiskers show, respectively, 25–75 quartiles, median, and minimum and maximum values of data collected from three samples per condition. **i** Fraction of dimerized molecules as detected using DNA-PAINT-SPT during ligand-titration experiments. Fitting the data with an analytical homodimerization model results in dissociation constants $K_X = 5.9 \pm 0.5 \times 10^{-3}\,\mu m^{-2}$, $K_B = 33 \pm 5$ pM. Insets: Trajectories of detected dimers for selected ligand concentrations, collected during 40 s measurements. Fields of view are 70 μm × 70 μm. Error bars denote mean and standard deviation of data collected on five fields of view per condition, samples were prepared independently as duplicates. Source data are provided.

binding proteins (FKBPs), with respectively complementary GA- and GT-imager strands in solution: 0.11 μm⁻²; localization density of poly(TC)- and poly(AC)-labeled FKBP proteins, with non-complementary GT- and GA-imager strands in solution: 0.00064 μm⁻² and 0.0013 μm⁻², respectively; Fig. 1b). This ensured that we could use the co-diffusion of single-molecules in two different color channels as a means of detecting their molecular interaction. We employed our dual-color DNA-PAINT-SPT method to investigate ligand-induced FKBP homodimerization by anchoring His-tagged FKBP to nickel-nitrilotriacetic acid (NTA-Ni)-containing SLBs. This system enables the study of 2D protein-protein interaction, preserving diffusive mobility and rotational freedom of the protein. We labeled the proteins via their SNAPtag using two orthogonal sets of benzylguanine (BG)-modified DNA docking strands, and reconstituted the labeled proteins in equimolar amounts. When we added the respective imager strands without the dimerization agent, no co-diffusion was detected (Supplementary Fig. 2), suggesting that the DNA-PAINT-SPT label itself does not introduce interactions. When adding the dimerization agent AP20187 in solution[13,14] (Fig. 1c), we reliably detected co-diffusion events and hence protein dimerization (Fig. 1d, e, Supplementary Fig. 3, Supplementary Fig. 4, Supplementary Movie 1, Supplementary Movie 2) using a total internal reflection fluorescence

microscope (TIRFM). Individual dimers could routinely be followed for tens of seconds, often even for several minutes (Fig. 1f, g; Supplementary Fig. 5, Supplementary Movie 3), in stark contrast to various single-dye labeled controls, including state-of-the-art fluorophores developed for single-molecule imaging[15] (Fig. 1f, g; Supplementary Fig. 6). Similarly, the number of observable DNA-PAINT labeled proteins remained more stable than the single-dye controls: not only does the number of detected dimers rapidly decrease over time for single-dye labeling, but also the apparent dimer lifetime is shortened as a consequence, resulting in a systematic underestimation of dimer stability (medians of co-diffusion durations measured with DNA-PAINT and single-dye labeling: $T_{DNA-PAINT-SPT} = 24.2 \pm 3.8$ s, and $T_{single-dye} = 4.1 \pm 1.3$ s; Fig. 1g). As a result of the frictional drag incurred by the increased membrane footprint of the dimers, we observed slowing down of diffusion of dimerized proteins by 28 % (Fig. 1h).

## Obtaining 2D dissociation constants for homodimerization from ligand-titration experiments

When titrating the ligand from low to increasingly higher concentrations, we could follow the onset of dimerization at around 1 pM, detected a maximal interaction level at concentrations around 100 pM and eventually the diminishing of dimerization due to the saturation of

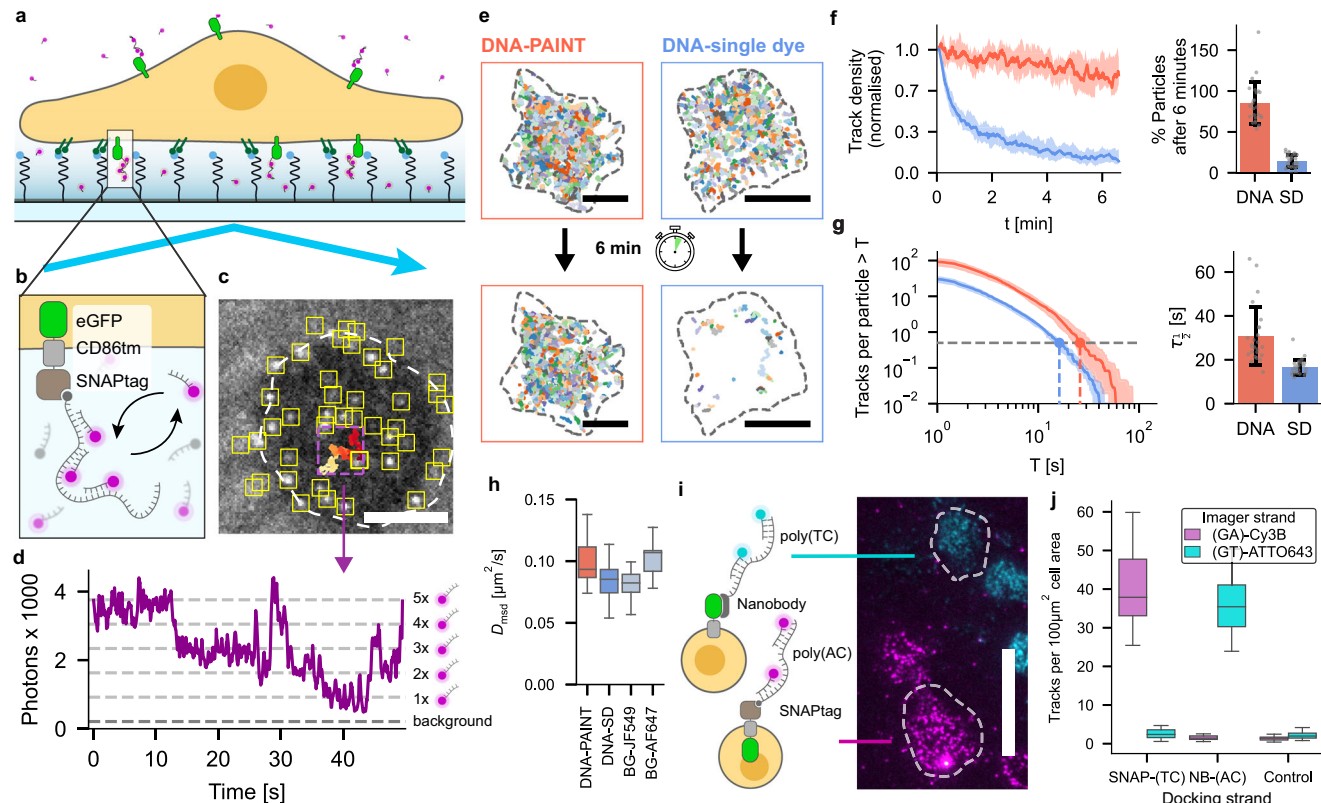

**Fig. 2 | DNA-PAINT-SPT enables extended tracking of individual membrane proteins on live cells. a** Schematic of DNA-PAINT-SPT live cell experiment: cells rest on PEG-cushioned SLBs decorated with RGD peptides for optimal cell attachment, imager access and surface passivation. **b** Membrane proteins of interest are labeled with an oligonucleotide (docking strand) using protein tags (SNAPtag) or nanobodies, allowing continuous binding and unbinding of fluorescently-labeled, complementary imager strands from solution. **c** Video frame of TIRFM video showing single-molecule trajectories of DNA-PAINT labeled membrane proteins expressed on a Jurkat T cell. Localized molecules (yellow boxes) and trajectory of an individual molecule (magenta box, trajectory is color-coded in time from red to yellow). This experiment was repeated independently five times with similar results. Scale bar: 5 μm. **d** Intensity of single molecule trajectory displayed in Fig. 2c, (magenta box) over time. Dashed dark gray line is the background level, light gray lines are guides to the eye indicating multiples of the lowest photon intensity, corresponding to the number of imager strands bound to the docking strand. **e** Membrane protein (SNAPtag-CD86tm-FKBP-GFP) trajectories collected during the first (top) and last (bottom) 20 s of a 6 min recording; Left: membrane proteins labeled with DNA-PAINT (BG-DNA, nine repeats of eight nucleotides complementary to Cy3B-labeled imager strand). Right: membrane proteins labeled with DNA single-dye control (BG-DNA, 35 nucleotides complementary to Cy3B-labeled imager strand). Scale bar: 10 μm. **f** Number of trajectories per frame on individual cells labeled with DNA-PAINT (red, $n_{cells}$ = 25) or single dyes (blue, $n_{cells}$ = 32),

normalised to initial number of trajectories. **g** Single-molecule trajectories on individual cells labeled with DNA-PAINT (red, $n_{cells}$ = 25) or single dyes (blue, $n_{cells}$ = 32) with a duration longer than $T$, normalised to the initial number of trajectories per cells. In **f** and **g**, curves represent the median with shaded areas indicating the 25–75 quartiles of data collected from three samples per condition. Bars and error bars denote mean and standard deviation. **h** Diffusion constants of SNAPtag-CD86tm-FKBP-GFP cell membrane proteins labeled with DNA-PAINT ($n_{cells}$ = 28), DNA single-dye control ($n_{cells}$ = 43) and organic fluorophores JF549 ($n_{cells}$ = 21) and AF647 ($n_{cells}$ = 7). Boxes, line and whiskers show, respectively, 25–75 quartiles, median, and minimum and maximum values of data. **i** Cells expressing either GFP-CD86tm-FKBP-HALOtag (cyan) or SNAPtag-CD86tm-FKBP-GFP (magenta) fusion proteins labeled with GFP-nanobody-conjugated docking strands (nanobody-DNA, nine repeats of eight nucleotides complementary to Cy3B-labeled imager strand) or with BG-DNA (BG-DNA, nine repeats of eight nucleotides complementary to ATTO643-labeled imager strand). Scale bar: 20 μm. **j** Density of tracks detected on cells expressing extracelullar SNAPtag- ($n_{cells}$ = 38) or GFP-fusion proteins ($n_{cells}$ = 35) and non-transfected controls ($n_{cells}$ = 30), when labeled in parallel with BG- or nanobody-conjugated docking strands with orthogonal sequences and their respective imager strands. Boxes, line and whiskers show, respectively, 25–75 quartiles, median, and minimum and maximum values of data. Source data are provided.

monomers with ligands at >10 nM (Fig. 1i, Supplementary Fig. 3). From these measurements, we were able to extract 2D dissociation constants $K_B$ (ligand from solution binding to FKBP monomer; [M]) and $K_X$ (cross-linking of a ligand-bound monomer with a free monomer; [μm⁻²]), by fitting an analytical homodimerization model to our DNA-PAINT single-molecule data ($K_B$ = 0.85 ± 0.17 nM, $K_X$ = 2.6 ± 0.2 × 10⁻² μm⁻²; coefficient of determination $R^2$ = 0.9914; Fig. 1i, Supplementary Note 2, ref. 16). Inducing dimerization with an anti-SNAPtag antibody instead of the AP20187 ligand was reflected in a two-fold reduction of diffusion upon dimerization, compared to AP20187-induced dimers (see Supplementary Fig. 7, Supplementary Fig. 8) and larger dissociation constants ($K_B$ = 136 ± 41 nM, $K_X$ = 0.12 ± 0.02 μm⁻²; Supplementary Fig. 9, Supplementary Fig. 10), in line with the expectations from solution kinetics predicting weaker affinities for the antibody.

## Orthogonal DNA-PAINT-SPT of two membrane proteins in live cells using SNAPligand- and nanobody-modified docking strands

Having developed dual-color DNA-PAINT-SPT, we next sought to establish DNA-PAINT-SPT for live cell SPT experiments. Since DNA-PAINT-SPT relies on the diffusive exchange of fluorescent imager strands from solution, a surface-restricted excitation geometry (i.e., TIRFM) is required to suppress the background signal from free imager strands. However, the implementation of TIRFM for DNA-PAINT-SPT on live cells is not trivial: the glass surface properties need to be tuned to facilitate cell adhesion while allowing imagers to diffuse underneath the cell (Fig. 2a). At the same time, unspecific binding of free imagers has to be minimal. Out of all passivation methods screened, we found that SLBs containing lipids modified with an integrin-recognition

peptide (DSPE-PEG2000-RGD) suppressed nonspecific binding the best while promoting cell attachment and imager strand diffusion underneath the cells (Fig. 2a and Supplementary Note 3, Supplementary Fig. 11, Supplementary Movie 4). During the live cell imaging time scales typical for SPT experiments (minutes to max. two hours), we observed no significant effects on cell viability or morphology beyond the differences in cell adhesion to the surfaces. For Cy3B-conjugated imager strands, passivation using a non-covalent PLL-PEG/PLL-PEG-RGD coating was also sufficient (Supplementary Note 3). After surface optimisation, we labeled a model transmembrane protein (SNAP-CD86tm-FKBP-GFP) via its extracellular SNAPtag with a BG-DNA docking strand for DNA-PAINT-SPT (Fig. 2b): after addition of complementary imager strands carrying Cy3B fluorophores, individual membrane proteins on the cell surface were visible as diffusing bright fluorescent spots with step-wise fluctuating intensity, as expected from the continuous binding and unbinding of imager strands (Fig. 2b–d; Supplementary Movie 5). We note cells appear as dark shadows surrounded by elevated background (Fig. 2c) in our TIRFM videos, indicating that diffusion of imager strands into the space between cells and glass coverslips is restricted. However, using a 75 nucleotides DNA docking strand and 40 nM of imager strands, we could achieve continuous exchange as indicated by the step-wise intensity fluctuations (Fig. 2d). Comparing DNA-PAINT-SPT on these cells to single-dye controls, we found that the number of observable DNA-PAINT labeled membrane proteins remained more stable (>85% after 6 min) compared to the single-dye control, where this number decreased to less than a fifth of the initial value (<15% after 6 min), with most of the remaining observable molecules diffusing in from the cell boundaries (Fig. 2e, f, Supplementary Fig. 12). As a combined measure for average trajectory length and number, we plot the number of tracks that are longer than a given threshold time $T$, normalised to the number of molecules initially detected per cell, for DNA-PAINT and single-dye labeled membrane proteins (Fig. 2g, Supplementary Fig. 13). This shows that also for live cell imaging, DNA-PAINT-SPT not only keeps the number of observable molecules constant for long durations (>6 min), but also increases the duration of individual trajectories (Cy3B-DNA-PAINT-SPT: $\tau_{1/2} = 31 \pm 13s$; Cy3B-single-dye: $\tau_{1/2} = 17 \pm 3s$).

The diffusion constants for both DNA-PAINT and single-dye DNA labeled proteins were similar to the direct labeling of the SNAPtag with BG-conjugated organic fluorophores (DNA-PAINT with Cy3B-labeled imager strands: $0.093 \pm 0.017 \ \mu m^2/s$, single-dye DNA labeled with Cy3B: $0.085 \pm 0.014 \ \mu m^2/s$, BG-JF549: $0.082 \pm 0.020 \ \mu m^2/s$, BG-AF647: $0.107 \pm 0.016 \ \mu m^2/s$; Fig. 2h, Supplementary Fig. 14). This suggests that DNA-PAINT-labeling per se influences diffusion to a lesser extent than the choice of fluorophore[17]. We also note that we did not observe any exclusion effects of DNA-labeled proteins from cell-surface contacts (Fig. 2c, Supplementary Fig. 15, Supplementary Fig. 16). Using dual-color DNA-PAINT-SPT, we were able to track FKBP-dimers on live cells for minutes, a substantial improvement over interaction tracking durations of mere seconds that are typically attainable with conventional single-dye labeling (see Supplementary Fig. 17, Supplementary Movie 6). Notably, DNA-PAINT-SPT worked equally well with a second, non-covalent labeling approach, using a docking strand conjugated to an antiGFP-nanobody (Fig. 2b and i, j). Thus, we can use two orthogonal labeling approaches (BG-conjugated docking strands or antiGFP-nanobodies) in combination with the orthogonal docking-imager pairs for dual-color DNA-PAINT-SPT on live cells (Fig. 2i, j; Supplementary Movie 7).

## Discussion

In this work we present DNA-PAINT-SPT as a promising technique for simultaneous dual-color tracking of proteins on SLBs and live cells. We developed docking strands with orthogonal sequences and show that our method outperforms current state-of-the-art labeling in both color channels. The marked improvement in trajectory lengths makes DNA-PAINT-SPT an ideal tool to study 2D binding kinetics quantitatively,

allowing for the reliably extraction of 2D dissociation constants from single-molecule measurements using an analytical dimerization model. We quantified 2D-$K_D$ constants of two dimerization agents (AP20187 and monoclonal anti-SNAPtag antibodies) in our FKBP protein homodimer-system, showing that we can clearly differentiate between the two using our DNA-PAINT-SPT approach. Moreover, since in both cases, dimerization was induced by a soluble ligand, we could also verify that the DNA-PAINT-SPT labels do not induce interactions or crosslinking by themselves. The absence of these artifacts and the convenient one-to-one targeting of molecules via standard tagging approaches make DNA-PAINT-SPT superior over other photostable labeling approaches, such as quantum dots or gold particles[3]. Protein-protein interactions in membranes are vital to many aspects of cellular function, including cellular signalling. Hence, experimental techniques that can measure these interactions in situ are highly desirable[18]. We note that in contrast to DNA-PAINT-SPT, most traditional affinity measurement techniques, such as surface plasmon resonance, only give 3D-$K_D$ constants and on-rates that cannot easily be converted into their 2D counterparts which are the relevant measures of protein-protein interactions in membranes[16,18–20].

In a next step, we established dual-color DNA-PAINT-SPT on live cells and for labeling membrane proteins with two orthogonal approaches; either via an extracellular SNAPtag that could bind benzylguanine-conjugated docking strands, or with an extracellular GFP tag that was labeled by docking strands conjugated to antiGFP-nanobodies. These orthogonal tagging methods allow the labeling of two sets of proteins simultaneously and specifically with different colors. We expect DNA-PAINT-SPT to work with a wide range of other common tagging approaches in addition to those tested in this study[21–23]. While we carefully examined the orthogonality of our DNA-PAINT-SPT labels and employed widely-used tags that have been shown in many cases not to interfere with receptor function[24–27], it is crucial to validate experimentally that the chosen labeling method does not significantly affect the biological function of the receptors under study. To minimize the impact of the label on the protein function and interactions, smaller peptide tags such as the ALFA-tag[22] or the incorporation of unnatural amino acids to enable bio-orthogonal attachment of DNA-docking strands via click chemistry[28] could be employed. For reducing the background and unspecific binding of free imager strands, we found bilayers decorated with PEGylated lipids conjugated to integrin-ligands to provide a highly-passivating, cell-attachment promoting surface. As an alternative, we demonstrated that reconstituting the adhesion protein ICAM on nickelated bilayers yields comparable surface passivation and cell attachment properties. Since the vast majority of cell lines express integrins, we expect this method to be widely applicable[29–31]. Moreover, reconstituting other adhesion-promoting proteins on bilayers matched to the cell type of interest would likely yield similarly suitable adhesion and passivation properties[32].

As for any DNA-PAINT method, a major limitation of DNA-PAINT-SPT is that it cannot be applied to image intracellular targets in live cells. However, we believe that the effort of developing better tools for extracellular SPT is well invested, given the high importance of cell membrane protein dynamics and interactions in cellular signaling and homeostasis. Being able to follow molecules for longer times without significant photobleaching will allow researchers to access biological processes happening at longer timescales and the largely-increased number of trajectories improves statistical certainty for the detection of rare events.

In summary, DNA-PAINT-SPT is a promising technique for quantitative SPT of membrane proteins on both SLBs and live cell membranes while being simple to implement by the single-molecule community: its use of standard protein tags and the commercial availability of a wide range of DNA modifications and fluorophores make it a versatile method. In the future, further improvements of

organic dyes[7] and new ways of reducing the background signal in DNA-PAINT experiments in general[33,34] will directly benefit DNA-PAINT-SPT and allow researchers to visualise protein function in membranes with even higher spatiotemporal resolution.

## Methods

### Materials

Chemicals and materials used were HEPES (H3375, Sigma-Aldrich), sodium chloride (310166, Sigma-Aldrich), magnesium chloride (M8266, Sigma-Aldrich), phosphate-buffered saline tablets (P4417, Sigma-Aldrich), pyranose oxidase (P4234, Sigma-Aldrich), catalase (C40, Sigma-Aldrich), Trolox (238813, Sigma-Aldrich), BSA (A9418, Sigma-Aldrich), Uvasol chloroform (1.02447, Sigma-Aldrich), Uvasol methanol (1.06002, Sigma-Aldrich), sulfuric acid (258105, Sigma-Aldrich), hydrogen peroxide (216763, Sigma-Aldrich), fibronectin (F0895, Sigma-Aldrich), PLL (P4707, Sigma-Aldrich), PLL-PEG-RGD (PLL(20)-g[3.5]- PEG(2)/PEG(3.5)-RGD, SuSoS), His-tagged ICAM-1 (IC1-H52H5, ACROBiosystems), B/B homodimerizer (AP20187, Takara Bio), dibenzocyclooctyne-PEG4-maleimide (760676, Sigma-Aldrich), SNAP/CLIP-tag monoclonal antibody (6F9, clone 6F9, Chromotek), FluoTag-Q anti-GFP single-domain antibody (N0305, clone 1H1, Nanotag Biotechnologies), SNAP-Surface Alexa Fluor 647 (S9136S, New England Biolabs), SNAP-Surface ATTO488 (S9124S, New England Biolabs), Tetraspeck Microspheres 0.2 µm (T7280, ThermoFisher Scientific). SNAPtag-ligand Janelia Fluor dyes (BG-JF549i, BG-JF646, BG-JFX650) were kindly provided by Luke Lavis (Janelia Labs, HHMI). Lipids used were DOPC (850375, Avanti), DGS-NTA-Ni (790404, Avanti), DSPE-RGD (870295, Avanti), DSPE-PEG-cRGDyk (LP096262-2K, Biopharma PEG) and DOPE-ATTO390 (390-161, ATTO-TEC). All DNA oligonucleotides were obtained HPLC-purified from Eurofins, except for BG-modified docking strands (Biomers) and azide-modified docking strands (Metabion) used for nanobody-DNA conjugation. Cell biology media and supplements used were DMEM without Phenol Red (12-917F), DMEM with Phenol Red (12-604F, Lonza), RPMI 1640 with Phenol Red (L0500-500, biowest), RPMI 1640 without Phenol Red (11835-030, gibco), Fluorobrite DMEM (A18967-01, gibco), PenStrep (15140122, Sigma-Aldrich), Na-Pyruvate (BE13-115E, Lonza), L-Glutamine (25030-024, gibco), FBS (A3160802, gibco), Ultramem (BE12-743F, Lonza), GeneJuice (70967, Sigma-Aldrich).

### Molecular biology

*pHR-SNAP-CD86-mOrange-FKBP* plasmid was a kind gift from Ricardo A. Fernandes. *Tet-pLKO-puro* backbone for tetracycline-inducible was obtained from Addgene (accession number #21915). For live cell imaging, *pHR-SNAP-CD86-eGFP-FKBP*, *pHR-eGFP-CD86-HaloTag-FKBP* and *Tet-pLKO-puro SNAP-CD86-eGFP-FKBP(f36v)* were created using Gibson assembly. For AP20187-induced homodimerization of FKBP, we introduced a point-mutation (FKBP[f36v]) following previously published protocols[14,35] and created *pET30-10His-FKBP(f36v)-SNAP* using Gibson assembly. Plasmids were deposited at Addgene (accession numbers #200280-#200283).

### Cell biology

Jurkat cells were cultured in RPMI with Phenol Red, supplemented with 10% FBS, 1% PenStrep and 1% Na-Pyruvate. *Tet-pLKO-puro*-transduced Jurkat cells were cultured without tetracycline present in the medium, as the remaining leaky expression was sufficient for single-molecule experiments. HEK cells for lentivirus production were cultured in DMEM with Phenol Red, supplemented with 10% FBS, 1% PenStrep and 1% Glutamax. Phenol-Red free media were used for seeding Jurkat cells into 6-well plates before transfection and labeling.

### Transduction of Jurkat T cells

Jurkat cells were transduced using lentiviral transfection: To this end, HEK293T cells were first transfected with the *pHR* plasmid of interest, *psPax* and *pMD2G* using GeneJuice, following the manufacturer's protocol. Viral supernatant was collected after 72h and used for Jurkat transduction.

### Nanobody-DNA conjugation

FluoTag-Q anti-GFP single-domain antibody (nanobody) site-specific conjugation was performed via the single ectopic cysteine at the C-terminus, similarly to the method described previously[10]: Unconjugated nanobodies were thawed on ice, then 20-fold molar excess of bifunctional maleimide-Peg4-DBCO linker was added and reacted for 2h on ice. Unreacted linker was removed by buffer exchange using Amicon centrifugal filters (10,000 MWCO). Then, two equivalents of azide-functionalized DNA (5') were reacted with the DBCO-modified nanobodies overnight at 4 °C. Unconjugated protein and free DNA were removed by anion exchange chromatography using an ÄKTA pure system equipped with a Resource Q 1mL column.

### Recombinant protein expression

10His-FKBP[f36v]-SNAP was expressed in *E.coli* (Rosetta strain), purified via its His-tag using an ÄKTA pure system equipped with a HisTrap HP 1mL column, followed by size-exclusion chromatography using a Superdex 200 increase 10/300 GL column.

### Small-unilamellar vesicle generation

Lipids were dissolved in chloroform (to dissolve DSPE-PEG-RGD, 10% methanol were added, and the mixture was sonicated for 30 s in a bath sonicator) and stored in 1.5 mL glass vials with PTFE-lined caps at −20 °C. Lipid mixes were prepared from the stock solutions depending on the required bilayer composition (reconstitution experiments: 98.5% DOPC, 1% DGS-NTA-Ni, 0.5% DOPE-ATTO390; live cell experiments: 89.5% DOPC, 10% DSPE-PEG-RGD, 0.5% DOPE-ATTO390) and 1 mL were transferred to a 50 mL round bottom flask. By gentle swirling and nitrogen flow, the lipid was dried into a thin film onto the flask walls. Once dried, trace amounts of chloroform were removed by desiccating the flask for at least 2h protected from light. Afterwards, the dried lipid film was rehydrated in HBS (HEPES 40 mM, pH 7.6, NaCl 140 mM) at a concentration of 2 mg/mL, aliquoted and stored at -20 °C until further use. On the day of the experiment, aliquots were thawed and sonicated for 30 min in a bath sonicator to produce small-unilamellar vesicles (SUVs). SUVs were diluted towards 0.1 mg/mL and used on the same day.

### Preparation of imaging chambers

Coverslips with the dimensions 25 x 75 mm, 1.5H (10812, Ibidi) and 22 x 22 mm, 1.5H (631-0851, VWR) were piranha cleaned using $H_2SO_4$ and $H_2O_2$ in a 3:1 ratio. After 1 h, they were thoroughly rinsed with milliQ water and dried using nitrogen flow. Slides were then air-plasma cleaned for 10 min (Harrick Plasma Cleaner PDC-002-HPCE). Chambers were created either by adhering ibidi sticky-slide 8-well or 18-well chambers (80808 or 81818, ibidi) onto pre-treated glass coverslips, or glueing 0.5 mL Eppendorf tubes with the conical part cut off onto coverslips, using UV-curable optical adhesive (NOA68, Thorlabs) and a 36 W UV nail dryer (B00R4M0TI0, Nailstar). Immediately after plasma cleaning and chamber assembly, 50 µL of HBS were added to each chamber. Then, 50 µL of the respective SUV solution (0.1 mg/mL) were added and the samples were transferred to a moisturised box for 30 min. Chambers were washed with 2 mL HBS and bilayers were blocked using BSA (1% (w/v) in HBS) for 10 min. Chambers were washed again with 2 mL HBS and stored in a dark moisturised box until further use on the same day. For the screening of surface passivation methods, fibronectin (100 µg/mL, 1h incubation), PLL-PEG-RGD (0.8 mg/mL, 2h incubation), PLL (100 µg/µl, 1h incubation) and His-tagged ICAM-1 (20 nM, 1h incubation on SLB containing 1% DGS-NTA-Ni) were used.

## Protein labeling and reconstitution in SLBs

FKBP[f36v] was thawed on ice and centrifuged for 1h at 4 °C at 16100 x $g$. For dual-color experiments, supernatant (2.1 μM) was divided into two aliquots and incubated separately with the respective docking strand (3.6 μM) for 1h at room temperature. Protein-DNA was diluted in HBS with 0.1% BSA and 10 mM $MgCl_2$ and incubated on the prepared lipid bilayers for 1h in a dark moisturised box at room temperature. Chambers were washed with 2 mL HBS and imager strand solution containing 40 nM of the respective imager strands, 5 mM $MgCl_2$, 3.7U/mL pyranose oxidase, 200U/mL catalase, 0.8% glucose, 0.1% (w/v) BSA and 2 mM Trolox-Trolox-quinone (Trolox to Trolox-quinone ratio 10% to 20%, determined via NanoDrop 2000 absorption[36]), were added.

## Cell labeling and preparation for microscopy

Cells were incubated with BG-DNA docking strands (1 nM to 10 nM for Jurkat cell lines with *pHR*-promoter plasmids, 100 nM for Jurkat cell lines with tetracycline-inducible *Tet-pLKO-puro*-promoter plasmids), nanobody-docking strands (5 pM to 10 pM) or BG-fluorophores (1 nM to 10 nM) for 30 min at 37 °C, 5% $CO_2$. Then, cells were washed three times by centrifugation (5 min, 100 x $g$) and re-suspending in PBS. The final resuspension step was performed in PBS with 5 mM $MgCl_2$ and 0.1% (w/v) BSA and, if applicable, the respective imager strand concentration: DNA-PAINT-SPT experiments were performed with 40 nM imager strands; for DNA single-dye experiments, 100 pM of complementary fluorescently labeled strands were added and washed out after 10 min of incubation. Live cell measurements were limited to a maximum imaging duration of two hours after removing the cells from the incubator. Cells with labeling densities of around 0.1 μm$^{-2}$ were used for analysis.

## TIRF microscopy

Fluorescence imaging was performed on a custom-built microscope in an objective-type TIRF configuration with an oil-immersion objective (CFI Apochromat TIRF 60x, NA 1.49, Nikon) and a three-color detection scheme[37]. A pre-assembled laser combiner was used to provide four excitation wavelengths (C-FLEX laser combiner, Hübner Photonics; 405 nm 140 mW, 488 nm 200 mW, 561 nm 220 mW, 638 nm 195 mW). The excitation beam was delivered to the optical bench via a single-mode polarisation-maintaining fiber (kineFLEX-HPV-P-3-S-405..640-0.7-0.7-P0, Qioptiq). The laser light was re-collimated after the fiber using an achromatic doublet lens ($f = 50$ mm) and directed through an achromatic quarter-waveplate to ensure circular polarisation. The laser beam was spectrally cleaned using a quad-line bandpass (ZET405/488/561/640xv2, Chroma) and then transformed into a collimated flat-top profile using a refractive beam shaping device (piShaper 6_6_VIS, AdlOptica)[38]. The laser beam diameter was magnified by a factor of 2.5 using a telescope assembly ($f_1 = 100$ mm, $f_2 = -40$ mm). The laser light was focused onto the objective's back focal plane using an achromatic doublet lens ($f = 250$ mm). A stage (KMTS25E/M Motorised Translation Stage, Thorlabs) translated the excitation beam off-axis to switch between wide-field, HILO or TIRF imaging. A short penetration depth of the evanescent field was ensured by translating the excitation beam in the back focal plane to the maximum possible value without clipping the beam. The angle of incidence was determined using a sample with fluorescent dye in solution (1 μM Cy3B-conjugated DNA), as previously described[39]: a circular aperture was placed in the beam path and the lateral displacement of the illuminated circle was measured upon translating the sample along the $z$-axis. The excitation beam was directed towards the objective by a four-color notch dichroic mirror (ZT405/488/561/640rpc-UF2, Chroma). Fluorescence emission passing through this dichroic mirror was spectrally filtered with a quad-line notch filter (ZET405/488/561/640mv2, Chroma) and was directed through a tube lens (TTL200-A, Thorlabs). The dichroic mirror, the objective, the tube lens and the quad-band notch filter were all placed in a CNC-milled cube based on

the miCube design[40]. This block also supported a piezo stick-slip stage (SLS-5252, Smaract) to move the sample in *x-y-z*. The tube lens formed an image outside of the cube, where a custom-built slit aperture was used to crop the image horizontally to enable simultaneous three-color imaging. In a 4f-system ($f = 300$ mm), the fluorescence emission was split spectrally using two dichroic mirrors (ZT561rdc and ZT640rdc, Chroma), filtered using respective bandpass filters (525/30 Brightline, Semrock; ET595/50m, Chroma; 680/42 BrightLine, Semrock) and imaged on a sCMOS camera (primeBSI, Teledyne Photometrics). Individual lenses ($f = 300$ mm) on the imaging side of the 4f system ensured matching focal planes for all three channels. The imaging setup resulted in an effective pixel size of 108 nm. Focus stabilisation was achieved using a system based on the pgFocus device[41]. An infrared laser (CPS808S, Thorlabs) was attenuated using a neutral density filter (NE13A-B, Thorlabs), coupled into the excitation path using a long pass dichroic mirror (ZT775sp2-2p-UF1, Chroma) and focused onto the back focal plane of the objective using a $f = 500$ mm lens. Using a manual micrometer stage, the infrared laser was brought into total-internal reflection. The back reflection was filtered through a bandpass filter (FB800-40, Thorlabs) and focused ($f = 200$ mm) onto a linescan sensor (TSL1401, Parallax). A feedback loop with the piezo-driven stage moving the sample allowed for focus stabilisation throughout extended measurement durations. The setup was controlled using K2 v1.0 software developed by Marko Seynen (AMOLF, Software Engineering Department).

## Imaging conditions

Fluorescence microscopy data was recorded at room temperature (22 ± 1°C) with our custom-built setup operating in three-color simultaneous imaging mode. To this end, the sCMOS camera readout was cropped to 682 x 2048 pixels, providing a 682 x 682 readout for each channel. The camera was operated at 32.4 ms (in vitro experiments) or 72.4 ms (live cell experiments) exposure times, with frame rates of 25 or 12.5 per second, respectively. The read-out rate was set to 100 MHz and the dynamic range to 12 bit. We performed experiments at laser excitation powers in the range of 4 mW to 40 mW (measured just before the back focal plane of the objective), which translate to irradiances of 15 W/cm$^2$ to 150 W/cm$^2$ in our setup. Laser powers were kept as low as possible while still allowing for robust localization above background levels, resulting in a typical localization precision between 20 nm and 30 nm for all labels. The angle of incidence was determined as 76° as previously described[39], resulting in a calculated evanescent field penetration depth ranging from 68 nm (561 nm laser excitation) to 78 nm (638 nm laser excitation).

## Data analysis

Raw data were localized using *picasso*[42]. Optical distortions were determined using a calibration slide with 200 nm Tetraspeck multi-color fluorescently-labeled beads, and corrected using *naclib 1.1.0*[43]. Trajectories were reconstructed from localizations using *trackpy 0.5.0*[44] and *Swift 0.4.3*[45,46]. Trajectories with durations of less than 10 frames or diffusion constants smaller than 0.01 μm$^2$/s were rejected. For analysis of live cell data, regions of interest were selected in *Fiji 2.9.0*[47] based on the cell outline of a maximum intensity projection of the underlying video or based on the GFP signal of the cell. Colocalization was determined by calculating intramolecular distances between localized molecules in the different color channels, and pairs with distances below 300 nm were identified as colocalized and potentially interacting. With a localization precision ranging between 20 nm and 30 nm and a color channel registration error of 32 nm, the threshold of 300 nm resulted in an expected false negative rate of <0.001%[48]. Colocalization events were linked frame-by-frame, with gaps of maximum 6 frames closed. Resulting co-diffusion trajectories were considered as true interactions if their duration exceeded 10 frames (see Supplementary Fig. 4 for an illustration of the analysis

workflow). Integration of the different analysis packages, any further analysis and visualisation of data was performed with custom-written *Python*-code available at github.com/GanzingerLab/SPIT and Zenodo[49] (https://doi.org/10.5281/zenodo.8043562).

## Statistics and reproducibility

Conclusions were based on data sampled from several thousand individually localized and tracked single molecules: For in-vitro experiments, we performed automated single-molecule localization and frame-by-frame tracking to build up trajectories from 32 supported lipid bilayer specimen, thereby collecting individual data from thousands of molecules. For live cell imaging, we detected the outline of 259 cells, performed automated localization and frame-by-frame tracking to build up trajectories of thousands of molecules on the surface of the cells. Trajectories shorter than 10 frames (too few data points for meaningful statistical analysis), or with diffusion constants smaller than $0.01\,\mu m^2\,s^{-1}$ (immobile molecules) were rejected. Co-localization events shorter than 10 frames were rejected on the basis that random Brownian motion will also result in co-localization on these time scales. Replicates were used for error estimation. For both live cell and in vitro experiments, extra replicates were prepared in case bilayer preparation failed. This was the case in about 10% of the samples. The experiments were not randomized. No blinding was performed as data was fully analysed by an automated data analysis pipeline with identical parameter settings for datasets that were subsequently compared.

## Reporting summary

Further information on research design is available in the Nature Portfolio Reporting Summary linked to this article.

# Data availability

Source data and *Python* notebooks to reproduce all graphs are available at Zenodo[50] (https://doi.org/10.5281/zenodo.7290071). Due to the large file size, the raw data that support the findings of this study will be provided by the corresponding author upon request within 3 weeks.

# Code availability

Python code for the analysis of the data is available at the public repository github.com/GanzingerLab/SPIT and deposited at Zenodo[49] (https://doi.org/10.5281/zenodo.8043562).

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

## Acknowledgements
We thank Elia Escoffier for testing data analysis code, Elisa Teunissen for maintaining cell lines and performing protein purification and Manuel Reinhardt for expert advice in optimizing the interaction analysis via colocalization thresholds. We are grateful for support from AMOLF technical engineering departments. This publication is part of project number VI.Vidi.203.037 of the Talent Programme which is financed by the Dutch Research Council NWO (K.A.G.). C.Ni. and K.A.G. also acknowledge support by the WISE program of NWO.

## Author contributions
C. Ni. conceived and performed live cell and in vitro experiments, analyzed and interpreted data and wrote the manuscript. C.Ng. performed live cell and in vitro experiments. M.W.-H. performed initial in vitro experiments. C.Ni. and M.W.-H. wrote analysis code. J.S. and F.S. performed initial experiments. S.S. performed nanobody-DNA coupling and reviewed the manuscript. A.C. derived analytical homodimerization model. J.S., A.C., R.J. and P.S. interpreted data and reviewed the manuscript. K.A.G. conceived and supervised the study, interpreted data and wrote the manuscript.

## Competing interests
The authors declare no competing interests.
