## [Peer Review File · Nature Communications]

REVIEWER COMMENTS

Reviewer #1 (Remarks to the Author):

Niederauer et al. apply the DNA-PAINT approach previously developed by the authors labs (Stehr et al, Nat Commun 2021) to a dual-color experimentation and live cell imaging. While the method has certainly potential, it is rather artificial and inherently limited to extracellular proteins. The two cellular applications shown appear rather preliminary from a cell biological perspective. Overall, only few additional new data are presented and the advancement over their original publications appears to be rather limited. As such, the manuscript does not qualify for publication in Nature Communications in my view.

Reviewer #2 (Remarks to the Author):

In this manuscript, the authors report an interesting technique to track proteins in a single-molecule resolution with long lifetime. This feature is attained by elaborate design of two types of single-stranded DNAs (docking and imager strands), especially by continuous association/dissociation of short imager strands. As the technique itself was previously reported in the author's recent publication (Ref. 8), the novelty in this manuscript lies in application to dual-color imaging. The authors show that dual-color DNA-PAINT-SPT can monitor i) homodimerization events of FKBP on a lipid bilayer and ii) diffusion of two transmembrane artificial proteins as two distinct colors on live cells. The overall data are valuable together with the rigorous kinetic and imaging analyses. A revision according to the following comments is recommended.

1) While the authors use the dimerizable transmembrane artificial proteins (GFP-CD86tm-FKBP-HALotag and SNAPtag-CD86tm-FKBP-GFP) for live cell imaging, the authors only track the diffusion of the two protein species. The manuscript would be strengthened if the authors could characterize FKBP ligand-induced dimerization events in the dual-color DNA-PAINT-SPT live-cell imaging.

2) Although the authors do not demonstrate in this manuscript, the DNA-PAINT-SPT technique would be developed for kinetically characterizing native-receptor physiology such as ligand-induced dimerization in live cells. However, since the SNAP tag and GFP are relatively big, fusion of these protein tags to a receptor of interest would profoundly affect receptor conformation and overall physiology, which may even destroy ligand recognition and proper dimerization. The authors should discuss how to solve this problem and accurately measure physicochemical kinetics of native receptors.

Reviewer #3 (Remarks to the Author):

The article describes an advancement on previously published work by the same authors (e.g. <https://www.nature.com/articles/s41467-021-24223-4>) using a variant of DNA-PAINT with multiple imager-strand binding sites to enable single particle tracking with greatly reduced bleaching. The principle advance over previously published work would seem to be the extension to simultaneous dual-colour tracking. Although this extension is conceptually predictable from previous work, the implementation is well executed and the dimerisation assay application example is particularly elegant. It would likely be very hard to achieve similar quality results with alternative tracking methods.

Overall I can find very little to fault with the paper - like the previous single-colour implementation it is technically excellent. I was particularly impressed by the realistic assessment in the discussion of where the technique would likely be most useful, including the acknowledgement that it would likely not be suitable for use for targets in the interior of live cells. This is a very welcome departure from the tendency within the field to over-claim.

There are a few minor issues the authors might like to address in a revision:

Data Analysis methods (p7 ll 277-279)

"Colocalization was determined by calculating intramolecular distances between localized molecules in the different color channels. Pairs with distances below 300 nm were marked as interaction candidates. Co-diffusing pairs that colocalize for a duration of at least 10 frames were marked as interacting, and gaps of up to 6 frames were closed."

This is slightly ambiguous as to whether pairs with distances <300 nm are deemed to colocalize, and hence whether the 300 nm distance is used in the test for interaction, or whether the 300 nm distance was simply used to find candidates and colocalisation/interaction tested on a different length scale.

If 300 nm is the interaction test length, it seems a bit long given a stated localisation precision of 20-30 nm, and suggests a missed opportunity here. This is probably out of scope for this paper, but in a cell membrane (rather than a SBL) being within 300 nm might simply indicate localization to the same domain, rather than interaction. I feel that a slightly more sophisticated test for interaction (correlation between movements?, or even a tighter distance threshold) might allow the distinction between domain-localisation and interaction to be made. I think it's reasonable to leave more complex analysis for cases / applications where it is required, but would like to see the interaction test length stated more clearly.

Imaging conditions (p7 l270)

"The angle of incidence was 76°, resulting in an evanescent field penetration depth in the range of 68 nm (561 nm laser excitation) to 78 nm (638 nm laser excitation)."

It is unclear if this penetration depth is a theoretical calculation or a measurement. If a calculation, changing the text to "resulting in a calculated evanescent field depth ..." would make it unambiguous. Whilst seemingly semantic, this distinction is important as small errors in angle measurement and/or beam divergence can have a big impact on the actual penetration depth.

Reviewer #4 (Remarks to the Author):

The authors characterize a useful method for dual-color single particle tracking by extending on a previously published DNA-PAINT-SPT approach, where prolonged imaging with reduced photobleaching is possible as more than one imager strand exchanges with the docking strand. They introduce a new sequence motif allowing for orthogonal labeling, thus enabling tracking of two targets over longer periods of time. In addition, the authors optimize conditions for live cell measurements of membrane-associated proteins. Of particular interest, the method enables direct measurement of dissociation constants. The manuscript is well written and the method is well described and simple to implement and thus should prove very beneficial for studying biomolecule dynamics in membranes of living cells.

Before publication, few major and some minor points should be addressed:

major comments:

- The authors refer to "single-dyes" multiple times throughout the manuscript without specifying the exact dye. Clarification throughout the text is suggested.
- line 84: Multiple surface passivation strategies were tested but possible side effects on cell-growth, morphology or viability were left uncommented. Since the method is suggested to be used as a live-cell imaging method, please comment on your observations.

minor comments:

- line 43: "negligible crosstalk": provide precise value as it can hardly be estimated from graph
- line 46: "nickel-nitrilotriacetic acid(Ni-NTA)": provide description of what it's used for
- line 46: "SLBs ": Describe and define what it stands for

- caption Fig1a: "short": give precise value
- Fig1f: add Legend. Specify and define "SD"
- caption Fig1f: "single-dye": Define which dye is shown
- Fig1f: Inset: Provide some x-Axis reference. For example, the end value
- Fig1g: definition for "SD" missing
- caption Fig1g: Define which "single-dye" is shown

- line 55: "stable": is overstated. Fig1f shows a 2-fold reduction in Dimer density. Use for example "more stable than the comparison"
- line 68: "excellent": remove wording and provide an estimate for Goodness of fit
- line 71: "drastic": remove wording. Rather provide estimate e.g. "two-fold"
- line 80: "surface"  glass-surface
- line 85: "selected fluorophores". Which?
- line 89: "are"  were
- line 95: "stable"  "more stable compared to.." (stable is wrong as there is an obvious decrease)
- line 101: "constant"  not constant, there is a decrease
- line 102: "single dye": which?

- Fig2b: might be helpful to label all protein parts

- line 120: "dimerisation agents": name the two for better readability
- line 150: "easy": switch to for example simple or straightforward
- line 167: "DGS-NTA-Ni": Previous Ni-NTA definition is not used
- line 177: Describe Gibson-Assembly parts for reproducibility

- Please consider providing Plasmids on AddGene for the community.

Response to the reviewers

We would like to thank all reviewers for their thorough evaluations and for their positive feedback appreciating our efforts. We are particularly grateful for the critical remarks regarding the colocalization analysis and the limitations of the technique, as revisiting these points has allowed us to significantly strengthen the manuscript. The changes made will aid future users in their own implementations of the DNA-PAINT single-particle tracking (DNA-PAINT-SPT). Importantly, we now show additional data demonstrating that DNA-PAINT-SPT is outperforming traditional single-dye labeling for tracking dimers on live cells. We believe the revised manuscript demonstrates even more clearly the advantages of DNA-PAINT-SPT, and is more comprehensive and easy to read for other researchers. In the following, we address each comment by the reviewers individually and highlight the changes/additions in the manuscript.

Reviewer 1

Reviewer Point P 1.1 — Niederauer et al. apply the DNA-PAINT approach previously developed by the authors labs (Stehr et al, Nat Commun 2021) to a dual-color experimentation and live cell imaging. While the method has certainly potential, it is rather artificial and inherently limited to extracellular proteins. The two cellular applications shown appear rather preliminary from a cell biological perspective. Overall, only few additional new data are presented and the advancement over their original publications appears to be rather limited. As such, the manuscript does not qualify for publication in Nature Communications in my view.

Reply: We regret that the reviewer does not see the potential that all other reviewers (and many other researchers from the microscopy community who have seen this work at conferences) attest DNA-PAINT-SPT. While the advancement presented may seem small, all novelties (dual colour tracking, reliable 2D- K_D quantification, tracking on live cells...) are conceptually simple but hard to implement in practice and the results of years of optimisation and many experiments. This was also clearly appreciated by all three other reviewers.

Reviewer 2

Reviewer Point P 2.1 — While the authors use the dimerizable transmembrane artificial proteins (GFP-CD86tm-FKBP-HALOtag and SNAPtag-CD86tm-FKBP-GFP) for live cell imaging, the authors only track the diffusion of the two protein species. The manuscript would be strengthened if the authors could characterize FKBP ligand-induced dimerization events in the dual-color DNA-PAINT-SPT live-cell imaging.

Reply: We agree with the reviewer that examining dimerization in live cells using dual-color DNA-PAINT-SPT would indeed be a natural extension of our experiments. However, the experimental setup in our initial submission involved a cell line stably overexpressing FKBP proteins, with only a small fraction of the population being labeled to achieve single-molecule densities. Consequently, direct observation of dimerization becomes highly improbable, as the vast majority

of proteins remain unlabeled. Considering that only 1/1000 of all proteins are labeled, which is likely an overestimation, merely one in 2000 dimers would be observable with both proteins labeled in different colors.

In light of the reviewer’s comment, we addressed the issue of low detection probabilities by preparing a new cell line expressing SNAPtag-CD86tm-FKBP-GFP under an inducible promoter (pLKO-puro, Addgene Plasmid #200283). We induced FKBP expression at single-molecule tracking compatible levels and labeled the cells in saturating conditions. Subsequently, we tracked FKBP dimers using dual-color DNA-PAINT-SPT and compared it to conventional SNAPtag-ligand labeling. We are confident that our additional data clearly demonstrate the superior capabilities of DNA-PAINT-SPT in observing molecular interactions in live cells. The new data have been incorporated into the revised manuscript (p.5, l.115; Fig. S16, Fig. SV5):

Using dual-color DNA-PAINT-SPT, we were able to track FKBP-dimers on live-cells for tens of seconds, a substantial improvement over interaction tracking durations of mere seconds that are typically attainable with conventional single-dye labeling (see Fig. S16, Fig. SV5).

Figure S16. FKBP dimer trajectory durations on live-cells.

Histogram of detected co-diffusion durations of FKBP dimers on live cells, using Cy3B- and ATTO643 DNA-PAINT-SPT (red, $n = 17$ cells) or BG-JF5449i and BG-AF647 single-dye labeling (blue, $n = 15$ cells) with 10 nM dimerizing agent AP20187. Data collected during three-minute measurements on two identically prepared samples per labeling condition.

Figure SV5. Dual-color DNA-PAINT labeled FKBP dimer on Jurkat T cell.

TIRFM video (100 ms exposure time, replayed at 25 fps) of FKBP proteins diffusing on a Jurkat T cell membrane (green and magenta circles), labeled with dual-color ATTO643- and Cy3B-DNA-PAINT-SPT. A FKBP dimer observable for > 3 minutes is highlighted (blue trajectory). Scale bar: 5 μm .

Reviewer Point P 2.2 — Although the authors do not demonstrate in this manuscript, the DNA-PAINT-SPT technique would be developed for kinetically characterizing native-receptor physiology such as ligand-induced dimerization in live cells. However, since the SNAP tag and GFP are relatively big, fusion of these protein tags to a receptor of interest would profoundly affect receptor conformation and overall physiology, which may even destroy ligand recognition and proper dimerization. The authors should discuss how to solve this problem and accurately measure physicochemical kinetics of native receptors.

Reply: We agree with the reviewer that a main drawback of any single-particle tracking method is the requirement of labelling the proteins of interest (fluorescently, or with a scatterer), which will always alter the system studied.

In the future, the impact of the DNA-PAINT-SPT label could be further decreased by using short peptide tags instead of SNAP-tag or GFP (e.g. the ALFA-tag [5]) in combination with nanobodies, or by incorporating unnatural amino acids to the receptor of interest to which the DNA-docking strand can be bio-orthogonally added via e.g. click chemistry [11]. We now discuss these ideas and the need to test the effect of any label on biological function in the discussion of the revised manuscript.

However, we would also like to note that in many cases, the standard tags we used in this study are found to not affect the biological function of the receptors – this is one of the reasons that they are so heavily used in the field [20, 1, 16, 9]. Arguably, DNA-PAINT-SPT is even better than the state-of-the-art when it comes to labelling artifacts, as trajectories of the lengths achievable by DNA-PAINT-SPT could previously only be obtained using quantum dots as labels, which are much larger and more artifact-prone than DNA-PAINT-SPT labels, as monovalent functionalization of quantum dots is challenging, leading to artificial multimerization of receptors [21].

We now address this in more detail in the manuscript on p.5, l.146:

While we carefully examined the orthogonality of our DNA-PAINT-SPT labels and employed widely-used tags that have been shown in many cases not to interfere with receptor function [20, 1, 16, 9], it is crucial to validate experimentally that the chosen labeling method does not significantly affect the biological function of the receptors under study. To minimize the impact of the label on the protein function and interactions, smaller peptide tags such as the ALFA-tag [5] or the incorporation of unnatural amino acids to enable bio-orthogonal attachment of DNA-docking strands via click chemistry [11] could be employed.

Reviewer 3

Reviewer Point P 3.1 — Overall I can find very little to fault with the paper - like the previous single-colour implementation it is technically excellent. I was particularly impressed by the realistic assessment in the discussion of where the technique would likely be most useful, including the acknowledgement that it would likely not be suitable for use for targets in the interior of live cells. This is a very welcome departure from the tendency within the field to over-claim.

Reply: We thank the reviewer for these encouraging comments!

Reviewer Point P 3.2 — Data Analysis methods (p7 ll 277-279)

“Colocalization was determined by calculating intramolecular distances between localized molecules in the different color channels. Pairs with distances below 300 nm were marked as interaction candidates. Co-diffusing pairs that colocalize for a duration of at least 10 frames were marked as interacting, and gaps of up to 6 frames were closed.”

This is slightly ambiguous as to whether pairs with distances <300 nm are deemed to colocalize, and hence whether the 300 nm distance is used in the test for interaction, or whether the 300 nm distance was simply used to find candidates and colocalisation/interaction tested on a different length scale.

If 300 nm is the interaction test length, it seems a bit long given a stated localisation precision of 20-30 nm, and suggests a missed opportunity here. This is probably out of scope for this paper, but in a cell membrane (rather than a SBL) being within 300 nm might simply indicate localization to the same domain, rather than interaction. I feel that a slightly more sophisticated test for interaction (correlation between movements?, or even a tighter distance threshold) might allow the distinction between domain-localisation and interaction to be made. I think it’s reasonable to leave more complex analysis for cases / applications where it is required, but would like to see the interaction test length stated more clearly.

Reply: We thank the reviewer for their comment on the colocalization analysis and apologize for any confusion caused by the description provided in the paper. In this response, we aim to clarify our choice of colocalization threshold and its effect on our analysis. Briefly, the 300 nm colocalization threshold was used as a criterion to identify potential interaction candidates. Identified candidates were deemed interacting only if their colocalization duration exceeded 10 frames.

Figure 1 a) Expected probability density function $p(r)$ of pairwise distances of a sample of dimerizing molecules with a combined localization- and color registration precision of $\sigma = 44$ nm. The green area reflects the colocalizations that would be correctly identified (true positive, 93.9%) using a deliberately small threshold distance of 150 nm (dashed line) for illustration purposes. Correspondingly, the grey area represents the missed colocalizations (false negative, 6.1%). b) True positive rate for different threshold values, for otherwise identical parameters. c) Expected probability density function $p(r)$ of pairwise distances of a sample of non-interacting molecules with a surface density of $0.8 \mu\text{m}^{-2}$. The red hatched area reflects false positive colocalizations from molecules incidentally passing by each other.

Regarding the choice of the colocalization threshold, we want to point out that in addition to the localization precision, one must also consider the color channel registration error, and, depending on the size of the involved molecules, their intramolecular distance, when setting a threshold. Fig. 1a shows the expected probability-density function of pairwise distances of dimerizing molecules with localization precisions of 30 nm and a color registration error of 32 nm, and an intermolecular distance of 15 nm (calculations based on [15]). The localization- and registration errors lead to

a broad distribution of observed pairwise distances. The choice of the colocalization threshold therefore directly affects both the ratio of detected (true positive) and missed (false negative) colocalizations, but also the amount of false positive incidental colocalizations, as illustrated in Fig. 1b. We consulted the literature and found colocalization distance threshold values varying from 100 nm to 300 nm: $R = 100$ nm [19, 20], 150 nm [2, 6], 200 nm [14], 220 nm [7], 240 nm [17], 300 nm [8].

Figure 2 Pairwise distances of experimentally obtained colocalization events of a dimer sample in a dynamic monomer-dimer equilibrium (**a**), and a monomeric control (**b**), at the same surface density of about $0.08 \mu\text{m}^{-2}$. The left column shows raw colocalization events, with a distinct peak visible for the dimeric sample, indicating interaction. Discarding colocalizations with durations shorter than 10 frames removes incidental colocalizations.

The reviewer comment prompted us to re-evaluate the choice of our colocalization threshold. Using the parameters of our experimental setup, a lower threshold value of 200 nm would result in a false-negative rate of 0.7% (see Fig. 1b, calculated according to [15]). This is already low, but the threshold value of 300 nm that we used reduces the false-negative rate even further to 0.001%. While a larger colocalization threshold reduces the chance of missing true colocalizations, it also increases the chances of including pairs of molecules that are just passing by each other incidentally (as illustrated by 1b-c). Generally, these false positive colocalizations of non-interacting molecules are removed by performing a subsequent filtering step where particles co-diffusing for less than a certain number of frames are discarded. In the literature, similar to the colocalization threshold, there are different values mentioned for the minimum duration of co-diffusion that must be met to be considered as an interaction. In the studies that report on the minimum duration threshold, a value of $T = 10$ frames or less is used (10 frames [20, 19, 8], 3 frames [3]). The mathematical framework motivating this parameter is described in [15]. In our study, we used a threshold of 10 frames, which allowed us to keep the colocalization threshold high without negatively affecting our analysis. The effect of the duration filtering is best illustrated by its effect on the pairwise distance histograms. In Fig. 2, we plot the pairwise distances of detected colocalizations for a FKBP sample with 100 pM dimerizing agent (a), and a FKBP monomer control without dimerizing agent (b). As expected (see Fig. 1, [15]), a distinct peak is observed in the pairwise distance histogram of the dimer dataset but not in the monomer control. Applying the 10 frame duration threshold effectively removes incidental, false positive colocalizations in both datasets.

Figure 3 Detected colocalizations in a sample of reconstituted FKBP dimers (top, 100 pM dimerizing agent) and FKBP monomer control (bottom, no dimerizing agent). Left column shows colocalizations detected in the first 500 frames of the measurement, with colocalization thresholds R of 200 nm and 300 nm. Colocalizations are color-coded by the distance r of each individual colocalizing pair. Middle column shows remaining consecutive colocalizations, after tracking and discarding all trajectories shorter than 10 frames (400 ms). Right column shows co-diffusion trajectories with gaps of up to 6 frames closed, color-coded by their trajectory ID.

Lastly, to confirm the robustness of our analysis, we examined the impact of a lower colocalization threshold (200 nm versus 300 nm) on the analysis of our data. The results are presented in Fig. 3. The larger distance threshold does not introduce artefacts in neither the fraction of dimerized molecules, nor the mean duration of co-diffusion trajectories, beyond the expected small increase stemming from the actual increase in the true positive rate (the fraction of dimerized molecules increases from 25% for the 200 nm threshold to 26.7% for a 300 nm threshold; the mean

co-diffusion trajectory length increases by 2.5% from 214.7 for the 200 nm threshold to 220.0 frames for the 300 nm threshold).

Generally, we agree with the reviewer’s point, that in more complex environments, e.g. when it comes to discerning between physical interaction and merely localization in the same lipid domain, a close examination of the pairwise distance histogram, and potentially choosing a tighter colocalization threshold as a result, may be beneficial. We would like to thank the reviewer again for this comment, as it inspired us to take a closer look at the effects of the colocalization- and duration thresholds and what information one can extract from the pairwise distance histograms.

We describe the colocalization analysis procedure now in more detail (p.8, l.298), provide a reference describing the mathematical framework and a new figure (**Fig. S3**) illustrating the analysis steps (based on Fig. 2 and Fig. 3):

Colocalization was determined by calculating intramolecular distances between localized molecules in the different color channels, and pairs with distances below 300 nm were identified as colocalized and potentially interacting. With a localization precision ranging between 20 nm and 30 nm and a color channel registration error of 32 nm, the threshold of 300 nm resulted in an expected false negative rate of $< 0.001\%$ [15]. Colocalization events were linked frame-by-frame, with gaps of maximum 6 frames closed. Resulting co-diffusion trajectories were considered as true interactions if their duration exceeded 10 frames.

Figure S3. Interaction analysis framework.

Detected colocalizations in a sample of reconstituted FKBP dimers (**a**), top, 100 pM dimerizing agent) and FKBP monomer control (**b**), bottom, no dimerizing agent) during a 20 s measurement. Left column shows colocalizations, color-coded by the distance of each individual colocalizing pair, and a histogram of their pairwise distances. A colocalization threshold of 300 nm was used. Middle column shows remaining consecutive colocalizations and the respective pairwise distance histogram, after tracking and discarding all trajectories shorter than 10 frames (400 ms). Right column shows co-diffusion trajectories with gaps of up to 6 frames closed, color-coded by their trajectory ID. Scale bars are 10 μm .

Reviewer Point P 3.3 — Imaging conditions (p7 1270)

“The angle of incidence was 76° , resulting in an evanescent field penetration depth in the range of 68 nm (561 nm laser excitation) to 78 nm (638 nm laser excitation).”

It is unclear if this penetration depth is a theoretical calculation or a measurement. If a calculation, changing the text to “resulting in a calculated evanescent field depth ...” would make it unambiguous. Whilst seemingly semantic, this distinction is important as small errors in angle measurement and/or beam divergence can have a big impact on the actual penetration depth.

Reply: We thank the reviewer for pointing out this lack of clarity. The penetration depth was calculated based on the measurement of the angle of incidence performed as described in [12], Appendix C. We changed the text on p.8, 1.291:

The angle of incidence was determined as 76° as previously described [12], resulting in a calculated evanescent field penetration depth ranging from 68 nm (561 nm laser excitation) to 78 nm (638 nm laser excitation).

Reviewer 4

Reviewer Point P 4.1 — Of particular interest, the method enables direct measurement of dissociation constants. The manuscript is well written and the method is well described and simple to implement and thus should prove very beneficial for studying biomolecule dynamics in membranes of living cells.

Reply: We are grateful for the positive comments and appreciate the thorough review.

Reviewer Point P 4.2 — The authors refer to “single-dyes” multiple times throughout the manuscript without specifying the exact dye. Clarification throughout the text is suggested.

Reply: We fully agree and thank the referee for spotting this omission and updated the manuscript accordingly. We have provided specific details on the changes made in our response to the list of minor points (P4.8, P4.10, P4.11,P4.20).

Reviewer Point P 4.3 — line 84: Multiple surface passivation strategies were tested but possible side effects on cell-growth, morphology or viability were left uncommented. Since the method is suggested to be used as a live-cell imaging method, please comment on your observations.

Reply: We thank the reviewer for this comment. For the durations typically used in live-cell SPT experiments (minutes to max. 2 hours), we did not find any effects on cell viability or morphology beyond the differences in cell adhesion to the surfaces (significant cell growth/division is not expected on this time scale). Although not very commonly utilized in general, live-cell imaging using lipid bilayers functionalized with adhesion proteins or peptides as substrates is a fairly established method in biophysical research of cellular surface signaling[4, 18, 10]. We now discuss this on p.3, 1.89 of the revised manuscript:

During the live-cell imaging time scales typical for SPT experiments (minutes to max. two hours), we observed no significant effects on cell viability or morphology beyond the differences in cell adhesion to the surfaces.

Additionally, we now also state clearly in the method section that our observations were limited to imaging cells for maximally two hours (p.7, l.249):

Live-cell measurements were limited to a maximum imaging duration of two hours after removing the cells from the incubator.

Minor points

Reviewer Point P 4.4 — line 43: “negligible crosstalk”: provide precise value as it can hardly be estimated from graph

Reply: We now provide these values:

Localization densities of poly(TC)- and poly(AC)-labeled FKBP proteins, with complementary GA- and GT-imager strands in solution: $0.11 \mu\text{m}^{-2}$; localization density of poly(TC)- and poly(AC)-labeled FKBP proteins, with non-complementary GT- and GA-imager strands in solution: $0.00064 \mu\text{m}^{-2}$ and $0.0013 \mu\text{m}^{-2}$, respectively;

Reviewer Point P 4.5 — line 46: “nickel-nitrilotriacetic acid(Ni-NTA)”: provide description of what it’s used for

Reply: See reply to Reviewer Point P 4.6.

Reviewer Point P 4.6 — line 46: “SLBs “: Describe and define what it stands for

Reply: We now briefly mention the use of Ni-NTA for anchoring His-tagged proteins and define SLBs:

We employed our dual-color DNA-PAINT-SPT method to investigate ligand-induced FK506 binding protein (FKBP) homodimerization by anchoring His-tagged FKBP to nickel-nitrilotriacetic acid (Ni-NTA)-containing supported lipid bilayers (SLBs). This system enables the study of 2D protein-protein interaction, preserving diffusive mobility and rotational freedom of the protein.

Reviewer Point P 4.7 — caption Fig1a: “short”: give precise value

Reply: We replaced “short” with 8 nucleotides .

Reviewer Point P 4.8 — Fig1f: add Legend. Specify and define “SD”

Reply: We added a legend in Fig1f, replaced “SD” by ”single-dye” and specified the used dyes in the figure caption of Fig1f:

Mean number of dimers per frame observable using Cy3B- and ATTO643 DNA-PAINT-SPT (red) or Cy3B- and ATTO643 single-dye control (blue), normalised to their respective initial values.

Reviewer Point P 4.9 — Fig1f: Inset: Provide some x-Axis reference. For example, the end value

Reply: We now provide the relevant x-Axis reference in Fig1f, and moved the inset further up to avoid covering too much of the main figure. This required removing the x-Axis label (“condition”) of Fig1b, which we deemed unnecessary.

Reviewer Point P 4.10 — Fig1g: definition for “SD” missing

Reply: We replaced “SD” by ”single-dye” in Fig1g.

Reviewer Point P 4.11 — caption Fig1g: Define which “single-dye” is shown

Reply: We specified the used dyes in the figure caption of Fig1g:

“Histogram of interaction durations for ligand-induced dimerization as detected during six minute measurements, using Cy3B- and ATTO643 DNA-PAINT-SPT (red, mean = 24.2 ± 3.8 s) or Cy3B- and ATTO643 single-dye labeling (blue, mean = 3.3 ± 1.3 s).”

Reviewer Point P 4.12 — line 55: “stable”: is overstated. Fig1f shows a 2-fold reduction in Dimer density. Use for example “more stable than the comparison”

Reply: We changed the wording:

Similarly, the number of observable DNA-PAINT labeled proteins remained more stable than the single-dye controls.

Reviewer Point P 4.13 — line 68: “excellent”: remove wording and provide an estimate for Goodness of fit

Reply: We calculated the coefficient of determination (R-Squared value) from the fit residuals and error values, using the `r2_score` function of the Python package `sklearn.metrics`. We rephrased the sentence:

From these measurements, we were able to extract 2D dissociation constants K_B (ligand from solution binding to FKBP monomer; [M]) and K_X (cross-linking of a ligand-bound monomer with a free monomer; [μm^{-2}]), by fitting an analytical homodimerization model to our DNA-PAINT single-molecule data ($K_B = 0.85 \pm 0.17$ nM, $K_X = 2.6 \pm 0.2 \times 10^{-2} \mu\text{m}^{-2}$; coefficient of determination $R^2 = 0.9914$; Fig. 1i, Supplementary Note B, ¹⁶.)

Reviewer Point P 4.14 — line 71: “drastic”: remove wording. Rather provide estimate e.g. “two-fold”

Reply: We changed the sentence into [...] reflected in a two-fold reduction of diffusion upon dimerization, compared to AP20187-induced dimers.

Reviewer Point P 4.15 — line 80: “surface” → glass-surface

Reply: We changed ”surface” to glass-surface.

Reviewer Point P 4.16 — line 85: “selected fluorophores”. Which?

Reply: We now specify:

For Cy3B-conjugated imager strands, passivation using a non-covalent PLL-PEG/PLL-PEG-RGD coating was also sufficient.

Reviewer Point P 4.17 — line 89: “are” → were

Reply: We replaced “are” with were .

Reviewer Point P 4.18 — line 95: “stable” → “more stable compared to..” (stable is wrong as there is an obvious decrease)

Reply: We changed the sentence:

[...] we found that the number of observable DNA-PAINT labeled membrane proteins remained more stable (> 85% after six minutes) compared to the single-dye control, where this number decreased to less than a fifth of the initial value.

Reviewer Point P 4.19 — line 101: “constant” → not constant, there is a decrease

Reply: We corrected:

DNA-PAINT-SPT can maintain a high number of observable molecules.

Reviewer Point P 4.20 — line 102: “single dye”: which?

Reply: We specified the used dyes both for DNA-PAINT-SPT and single-dye:

Cy3B-DNA-PAINT-SPT: $\tau_{1/2} = 31 \pm 13$ s; Cy3B-single-dye: $\tau_{1/2} = 17 \pm 3$ s.

Reviewer Point P 4.21 — Fig2b: might be helpful to label all protein parts

Reply: We now label all protein parts in Fig2b.

Reviewer Point P 4.22 — line 120: “dimerisation agents”: name the two for better readability

Reply: We now name both dimerization agents:

We quantified $2D-K_D$ constants of two dimerization agents (AP20187 and monoclonal anti-SNAPtag antibody) in our FKBP protein homodimer-system.

Reviewer Point P 4.23 — line 150: “easy”: switch to for example simple or straightforward

Reply: We replaced ”easy” with simple .

Reviewer Point P 4.24 — - line 167: “DGS-NTA-Ni”: Previous Ni-NTA definition is not used

Reply: We changed the Ni-NTA definition to NTA-Ni, to conform with all NTA-Ni occurrences.

Reviewer Point P 4.25 — line 177: Describe Gibson-Assembly parts for reproducibility

Reply: Unfortunately, the technician who performed the cloning has left the institute and determining the exact steps taken has been challenging. However, we have sequenced the inserts and have decided to share them via Addgene (see reply to Reviewer Point P 4.26). We appreciate the concern for reproducibility and hope that this solution meets the expectations.

Reviewer Point P 4.26 — Please consider providing Plasmids on AddGene for the community.

Reply: We thank the referee for this valuable suggestion. We are very committed to open science and therefore followed the reviewer’s suggestion to share the plasmids on Addgene.

Addgene catalog number 200280: pHR-SNAP-CD86tm-eGFP-FKBP(f36v)

Addgene catalog number 200281: pHR-eGFP-CD86tm-HALOtag-FKBP(f36v)

Addgene catalog number 200282: pET30-10His-FKBP(f36v)-SNAP

Addgene catalog number 200283: pLKO-puro SNAP-CD86-eGFP-FKBP(f36v)

Other changes

Figure 1a

To enhance the visual representation of the DNA-PAINT-SPT principle, we have made several improvements to the schematic drawing presented in Fig1a:

- We have divided the DNA docking strand into visually distinct sections, each corresponding to an individual binding site for one imager strand. This allows for a clearer depiction of the repeating binding sites.
- To illustrate that there can be multiple imager strands bound to the docking strand at a given time, we have added imager strands to the schematic drawing.
- We replaced the abbreviations “bp” (basepairs) with “nt” (nucleotides), since the docking and imager strands are all single-stranded DNA by design.

Initial and revised versions of Fig. 1a.

Methods

In the description of the microscope, the reference to the lab’s GitHub page was replaced by a publication describing the microscope in detail [13].

Acknowledgements

In the acknowledgements section, another contribution was added: We thank [...] Manuel Reinhardt for expert advice in optimizing the interaction analysis via colocalization thresholds.

References

- [1] D. Calebiro et al. “Single-Molecule Analysis of Fluorescently Labeled G-protein-coupled Receptors Reveals Complexes with Distinct Dynamics and Organization”. In: *Proceedings of the National Academy of Sciences* 110.2 (Jan. 2013), pp. 743–748.
- [2] Antonios Drakopoulos et al. “Investigation of Inactive-State κ Opioid Receptor Homodimerization via Single-Molecule Microscopy Using New Antagonistic Fluorescent Probes”. In: *Journal of Medicinal Chemistry* 63.7 (Apr. 2020), pp. 3596–3609.
- [3] Paul D. Dunne et al. “DySCo: Quantitating Associations of Membrane Proteins Using Two-Color Single-Molecule Tracking”. In: *Biophysical Journal* 97.4 (Aug. 2009), pp. L5–L7.
- [4] Michael L. Dustin et al. “Supported Planar Bilayers for Study of the Immunological Synapse”. In: *Current Protocols in Immunology* 76.1 (2007), pp. 18.13.1–18.13.35.
- [5] Hansjörg Götzke et al. “The ALFA-tag Is a Highly Versatile Tool for Nanobody-Based Bioscience Applications”. In: *Nature Communications* 10.1 (Dec. 2019), p. 4403.
- [6] Jak Grimes et al. *Single-Molecule Analysis of Receptor- β -Arrestin Interactions in Living Cells*. Nov. 2022.
- [7] Rinshi S. Kasai et al. “The Class-A GPCR Dopamine D2 Receptor Forms Transient Dimers Stabilized by Agonists: Detection by Single-Molecule Tracking”. In: *Cell Biochemistry and Biophysics* 76.1-2 (June 2018), pp. 29–37.
- [8] Sarah L. Latty et al. “Referenced Single-Molecule Measurements Differentiate between GPCR Oligomerization States”. In: *Biophysical Journal* 109.9 (Nov. 2015), pp. 1798–1806.
- [9] Jan Möller et al. “Single-Molecule Analysis Reveals Agonist-Specific Dimer Formation of μ -Opioid Receptors”. In: *Nature Chemical Biology* 16.9 (Sept. 2020), pp. 946–954.
- [10] Pradeep M Nair et al. “Using Patterned Supported Lipid Membranes to Investigate the Role of Receptor Organization in Intercellular Signaling”. In: *Nature Protocols* 6.4 (Apr. 2011), pp. 523–539.
- [11] Franziska Neubert et al. “Bioorthogonal Click Chemistry Enables Site-specific Fluorescence Labeling of Functional NMDA Receptors for Super-Resolution Imaging”. In: *Angewandte Chemie International Edition* 57.50 (2018), pp. 16364–16369.
- [12] Christian Niederauer et al. “Direct Characterization of the Evanescent Field in Objective-Type Total Internal Reflection Fluorescence Microscopy”. In: *Optics Express* 26.16 (Aug. 2018), p. 20492.

- [13] Christian Niederauer et al. “The K2: Open-source Simultaneous Triple-Color TIRF Microscope for Live-Cell and Single-Molecule Imaging”. In: *HardwareX* 13 (Mar. 2023), e00404.
- [14] Verena Ruprecht, Mario Brameshuber, and Gerhard J. Schütz. “Two-Color Single Molecule Tracking Combined with Photobleaching for the Detection of Rare Molecular Interactions in Fluid Biomembranes”. In: *Soft Matter* 6.3 (2010), pp. 568–581.
- [15] Verena Ruprecht et al. “Measuring Colocalization by Dual Color Single Molecule Imaging”. In: *Advances in Planar Lipid Bilayers and Liposomes*. Vol. 12. Elsevier, 2010, pp. 21–40. ISBN: 978-0-12-381266-7.
- [16] Titiwat Sungkaworn et al. “Single-Molecule Imaging Reveals Receptor–G Protein Interactions at Cell Surface Hot Spots”. In: *Nature* 550.7677 (Oct. 2017), pp. 543–547.
- [17] Kenichi G.N. Suzuki et al. “Single-Molecule Imaging of Receptor–Receptor Interactions”. In: *Methods in Cell Biology*. Vol. 117. Elsevier, 2013, pp. 373–390. ISBN: 978-0-12-408143-7.
- [18] Marcus J. Taylor et al. “A DNA-Based T Cell Receptor Reveals a Role for Receptor Clustering in Ligand Discrimination”. In: *Cell* 169.1 (Mar. 2017), 108–119.e20.
- [19] Stephan Wilmes et al. “Competitive Binding of STATs to Receptor Phospho-Tyr Motifs Accounts for Altered Cytokine Responses”. In: *eLife* 10 (Apr. 2021), e66014.
- [20] Stephan Wilmes et al. “Receptor Dimerization Dynamics as a Regulatory Valve for Plasticity of Type I Interferon Signaling”. In: *Journal of Cell Biology* 209.4 (May 2015), pp. 579–593.
- [21] Yanqi Yu, Miao Li, and Yan Yu. “Tracking Single Molecules in Biomembranes: Is Seeing Always Believing?” In: *ACS Nano* (Oct. 2019).

REVIEWERS' COMMENTS

Reviewer #2 (Remarks to the Author):

The authors revised the manuscript in light of the reviewers' comments. Now the manuscript would be ready for publication.

Reviewer #3 (Remarks to the Author):

I am satisfied with the alterations made in the revised version, which seem to be thorough and thoughtful.

Reviewer #4 (Remarks to the Author):

The authors have adequately addressed all my points. I am happy to suggest this manuscript with beneficial method development for publication.